# Disinformation: A Bibliometric Review

**DOI:** 10.3390/ijerph192416849

**Published:** 2022-12-15

**Authors:** Shixiong Wang, Fangfang Su, Lu Ye, Yuan Jing

**Affiliations:** 1Library, Zhejiang Sci-Tech University, Hangzhou 310018, China; 2College of Economics and Management, Zhejiang Sci-Tech University, Hangzhou 310018, China

**Keywords:** disinformation, bibliometric analysis, keywords analysis, hot topics

## Abstract

Objectives: This paper aimed to provide a systematic review of relevant articles from the perspectives of literature distribution, research hotspots, and existing results to obtain the frontier directions in the field of disinformation. Methods: We analyzed disinformation publications published between 2002 and 2021 using bibliometric methods based on the Web of Science. There were 5666 papers analyzed using Derwent Data Analyzer (DDA). Results: The result shows that the USA was the most influential country in this area, while Ecker and Lewandowsky from the University of Western Australia published the largest volumes of papers. Keywords such as “social media”, “COVID-19”, and “vaccination” have gained immense popularity recently. Conclusions: We summarized four themes that are of the biggest concern to scholars: group heterogeneity of misinformation in memory, disinformation mechanism in social media, public health related to COVID-19, and application of big data technology in the infodemic. The future agenda of disinformation is summarized from three aspects: the mechanism of disinformation, social media users, and the application of algorithms. This work can be a meaningful resource for researchers’ study in the area of disinformation.

## 1. Introduction

Disinformation is non-accidentally misleading information [1]. It will do direct or indirect harm to people in venture capital [2,3], medical treatment [4,5,6,7], public opinion [8,9,10,11], and even political communication [12,13,14,15,16,17]. Consequently, it becomes extremely significant to review research relevant to disinformation. Prototypical varieties about disinformation are false information [18], misinformation [19], and information pollution [20]. To avoid a too broad or narrow definition, this paper focuses on the research process of disinformation.

Since the concept of “disinformation” was coined in the 1980s [21], many researchers have conducted rigorous scientific research from different perspectives. From a political perspective, disinformation has been instrumentalized, such as the function of misinformation during the election [22], and the dissemination of political disinformation on social media [23]. From an economic perspective, scholars discussed the negative effects of the diffusion of disinformation in the market, such as illegal huge profits [24], and deterioration of market liquidity [25]. From a social perspective, disinformation in social media has been a hot issue. Scholars used different criteria to screen the information on the Internet and found that a large amount of medical misinformation was misleading people to make wrong decisions. The misinformation was all about cancer [26], plastic surgery [27], vasectomy [28], prehospital care [29], and vaccines [30]. From a military perspective, researchers studied the application of disinformation in warfare in the context of the Cold War [21]. From an individual perspective, researchers studied the effects of automatic and intentional memory [31], representation ability [32], drinking [33], and other factors on people’s interference by misinformation through experimental methods in the 2000s. Several documents have provided suggestions for the prevention and control of disinformation. Some of these documents focused on theory: Huang et al. explored random and targeted immunization strategies and targeted immunization strategies [34]; Goslin et al. constructed website quality assessment models [35,36]; Nguyen et al. found influential nodes in the dissemination network to contain the spread of misinformation [37]. Some documents focused on practical applications: Fortinskyet et al. developed programs that provide services for recommending reliable resources [38]; Littman et al. put forward that education tools should be developed for providing accurate information [39]; Lin et al. selected a set of observers to spot disinformation in time before being disseminated widely [40]. Overall, the research on disinformation presents the characteristics of wide fields and diverse perspectives.

It is necessary to systematically analyze and summarize the research on disinformation from an overall perspective. Some scholars have made related reviews on the study of disinformation. Their work focused on the definition and identification of disinformation [41,42], the key research directions of disinformation [43,44], the introduction of different forms of disinformation [41,45], and the generalization of technologies to deal with disinformation [45]. These reviews summarized the theoretical knowledge of disinformation and synthesized the findings to some extent. However, systematic review tends to be dependent on qualitative analysis, which is inevitably affected by the intellectual background of researchers and causes deviations in conclusions [46].

Quantitative analysis methods can avoid the aforementioned bias. Bibliometric analysis and meta-analysis are two frequently used review alternatives that rely on quantitative methods [47]. Meta-analysis is more suitable for similar literature and open issues [48]. Based on the characteristics of the large-scale and variety of literature in this study [49], this paper endeavored to use bibliometric analysis to systematically summarize the whole picture of disinformation research from the number of papers, citation frequency, influencing factors, H-index, and other dimensions.

It is generally believed that the earliest bibliometric research started in the early 20th century. In the early stage of development, bibliometric methods were mostly used in medical fields [50]. Around the 1940s, the establishment and maturity of Brad Ford’s Law, Lotka’s Law, and Zipf’s Law laid a solid foundation for the development of bibliometrics [51,52,53]. Since then, this method has been widely used in natural sciences [54,55], subject areas like mathematics [56], physics [57], and chemistry [58].

There are a few papers that provide a bibliometric review of the concepts related to disinformation. They provide an overview in some perspectives. Lee used bibliometric methods to reveal the evolution of academic networks in the field of misinformation, but his study was limited to the period 2009–2018, which clearly did not reflect the latest developments [59]. Some scholars reviewed rumors, fake news, and information epidemics, but these are related concepts of disinformation, narrowing the scope of what disinformation can accommodate [60,61,62]. The review of Tito et al. and Yeung et al. limited disinformation to the social media [63] and medical fields [64], respectively, and although these two fields are the main sites of disinformation, they limit the grasp of the filed in its entirety. The review of Patra et al. summarized the existing results comprehensively to some extent, but rarely tapped into research hotspots and research trends that are directly related to disinformation [65]. Taken collectively, a comprehensive and objective analysis of the field of disinformation and an overview of the frontiers of development are lacking. This paper provides a systematic review as follows. First, the current status of research on this topic in various countries around the world is sorted out and analyzed from temporal and spatial perspectives. Second, based on the highly cited papers and hot papers, we provide an overview of the existing results in terms of theories, methods, and conclusions. Third, the hot topics in the field are explored, and the research frontiers are summarized based on the evolution of research objects and topics.

The remainder of this paper is divided into the following sections: methods and materials, results, discussion and expansion of new issues, and a summary of the full text.

## 2. Materials and Methods

The primary research method in this study is bibliometric analysis. It is not only a quantitative and qualitative analysis of the dissemination of scientific literature using mathematical and statistical methods [66], but also a powerful tool for summarizing existing knowledge structures and quantifying global scientific productivity in a specific field [67].

Bibliometric analysis usually consists of two parts: performance analysis and science mapping [68]. Performance analysis contributes to help discover emerging themes and recent advances in a field, the influence of leading scholars, and the impact of different journals and schools of thought [69]. This paper uses performance analysis to find leading countries/regions and journals, as well as prolific authors and institutions.

Bibliometric analysis relies on citation and co-citation analysis for quantitative review [70]. Citations indicate the use of a specific work by a citing scholar and reveal the value, importance, and influence of that work [69]. In this analysis, the most cited works are used to illuminate the theoretical underpinnings, methodologies, and key themes that drive the discipline in the field of disinformation.

Keyword co-occurrence analysis is a bibliometric method, which assumes that when two keywords appear in multiple articles at the same time, there must be some correlation between the concepts reflected [71]. It is considered appropriate to express central themes in some fields using keywords [72]. Based on the numerous examples of literature, keyword co-occurrence analysis is used to identify the central topics in disinformation.

With the advancement of bibliometric research, several analysis and visualization tools have been developed, such as VOSviewer, Citespace, and Derwent Data Analyzer (DDA). Among the tools, DDA is a more competitive software for cleaning, mining, and visualizing patent data and scientific literature. DDA can analyze and track scientific research activities in a particular research field. Hence, this study used the DDA 10 to present analyzing results in the form of charts and tables in this research.

Our work is based on the Science Citation Index-Expanded (SCI-E) and the Social Science Citation Index (SSCI) on the Web of Science. The retrieval formula is disinformation or misinformation or “fake news” or “infodemic” or “information pollut *” or deepfake * or “rumor propagation”. All the keywords were separated by an ‘or’ for further inclusivity [73]. The retrieval time is 10 January 2022. Results are restricted to the topic (title, keywords, and abstract) and time-restricted to 1990.

In addition, there are some variables and metrological indicators used in this paper as the following explanations:

TP: total publications.

TC: total citations of publications.

IF: impact factor of some journals in 2021.

ACPP: average citations per paper.

h-index: an indicator to describe the scientific productivity of researchers, representing that at most h papers have been cited at least h times.

## 3. Results

According to the retrieval criteria, a total of 7326 papers were obtained. Only papers in English belonging to “article” and “review” were screened. After removing papers from years with low number of papers (1947–2001), papers other than articles and reviews, papers published in languages other than English, and some non-closely related articles, 5666 related papers published from 2002 to 2021 were obtained. Three file types were involved: articles (*n* = 4976), reviews (*n* = 367), and other types (*n* = 323). The average citation frequency per paper for these papers was 50.97, and the total citation frequency was 299,681.

### 3.1. Countries/Regions Production and Collaboration

A total of 139 countries/regions were involved in the scientific research production of disinformation during 2002–2021. Figure 1 shows the growth of disinformation research papers and the top 20 countries/regions with the highest productivity. The number of papers related to disinformation has increased exponentially year by year. The USA, the UK, China, and Australia rank in the top four, according to citation frequency. Concretely, the number of disinformation-related papers published in the top 20 countries has increased over time, especially after 2019, when most countries such as the USA, the UK, and Australia achieved a surge. The articles published by the USA account for half of the top 20 countries, even amounting to 65.7% (2004). It must be stressed that although the number of papers published by the USA in the field has increased year by year, its percentage has had a significant downward trend.

Figure 2 shows the collaboration among the top 20 countries/regions in terms of the number of papers and the specific frequency of collaboration, with the nodes representing countries/regions, the size of a node representing the number of journal papers published in that country/region, and the straight line between two dots indicating the collaboration generated between the countries/regions.

Productive countries collaborate more with other countries, and they even have cooperation with each of the top 20 countries/regions, such as the USA, the UK, Australia, and China. The USA has closer collaborations with the UK (164 papers), Australia (102 articles), and China (102 articles). Except for the USA, the UK has the highest frequency of collaboration with Australia (85 papers) and The Netherlands (40 articles). The intensity of China’s partnerships with other countries is generally low. Besides the USA, China has the highest frequency of collaboration with Australia, but only 30 papers have been published. It should be underlined that some of the top 20 countries/regions have not jointly published papers yet, such as South Korea, New Zealand, and Poland.

### 3.2. The Most Attractive Journals

A total of 2116 journals published articles on disinformation research, among which 1252 publications published only one paper. The top 30 academic journals with the number of articles are listed in Table 1. These journals published 22.86% of the total number of papers. The top three academic journals are *APPLIED COGNITIVE PSYCHOLOGY*, *PLOS ONE*, and *JOURNAL OF MEDICAL INTERNET RESEARCH*. Specifically, there are 11 journals from the USA and eight from the UK, which accounts for 63.3% of the top 30. It should be noted that *MEDIA COMMUN-LISBON* comes from Portugal, which is not one of the top 20 high-yield countries/regions. In addition, these 30 journals generally have a high value of IF, with an average of 4.08. The journal with the highest IF is *P NATL ACAD SCI USA*, which published 38 papers. For further analysis, based on the characteristics of these journals’ publication volume over time, they can be divided into the following categories: firstly, the main journals represented by *APPLIED COGNITIVE PSYCHOLOGY*. Its relevant papers are produced every year from 2002 to 2021, and the number of papers published each year is relatively stable. Next are the rising stars represented by *IEEE ACCESS*, *SOCIAL MEDIA AND SOCIETY*, and *DIGITAL JOURNALISM*. They have a clear trend of growth in the number of papers published in recent years, especially *PLOS ONE* and *JOURNAL OF MEDICAL INTERNET RESEARCH* who have published the largest number of papers in the past two years. Finally, there are some journals with potential. Take *CONTRACEPTION, PROCEEDINGS OF THE NATIONAL ACADEMY OF SCIENCES OF THE UNITED STATES OF AME* as an example; it has been focusing on the field of disinformation, but the number of publications per year is not high.

### 3.3. Leading Authors

A total of 17,661 authors participates in the study of disinformation. Table 2 shows the top 30 high-yield authors. The authors are spread out across 10 countries/regions and 26 institutions. The most productive contributors in the field of disinformation are Ecker (TP: 37) and Lewandowsky (TP: 37) from the University of Western Australia, Australia, followed by Loftus (TP: 36) from Univ Calif Irvine in the USA. Among them, Lewandowsky has the highest value in the H-index and TC. His findings have been cited 3063 times in total, and the value of the H-index is up to 21, indicating that papers reported by him have a high impact in the field of disinformation. Although Reifler published only 11 papers relevant to disinformation, he ranks first in ACPP, demonstrating the high quality of his academic output. Additionally, consistent with the previous analysis, most of these authors come from the most productive countries/regions, like the USA (*n* = 10), Australia (*n* = 4), The Netherlands (*n* = 4), and the UK (*n* = 3). As one of the most productive countries in the field, China has only one author in the top 30 (Zhu from Jiangsu University).

In addition, we analyzed the cooperation among the top 30 high-yield authors. As shown in Figure 3, each rounded rectangle represents an author, the dot connected to the rounded rectangle represents the number of published papers, and the dot connected to the two rounded rectangles represents the number of papers jointly published by two authors.

The cooperation among high-yield authors is characterized by overall dispersion and localized intensity. In terms of breadth, there is more collaboration between Lewandowsky and the other top 30 authors, involving three countries: the UK, the USA, and Australia. Simultaneously, the relationship among Cook, Ecker, and van Der Linden is also typical multilateral cooperation. In terms of depth, the cooperative relationship between Pennycook and Rand is the most intimate, as they have cooperated 18 times. Other stable cooperative relationships include the cooperation between Bode and Varga, as well as the cooperation among Zollo, Quattrociocchi, and Scala. It is worth noting that Loftus and Zhu, who published many papers, have not cooperated with other top 30 most productive authors.

### 3.4. Leading Institutions

A total of 4888 institutions have participated in the study of disinformation. Table 3 shows the top 30 research institutions with the highest productivity, including 22 from the USA, three from the UK, two from Australia, and one each from Singapore, Canada, and The Netherlands. Among them, the total number of papers published by Univ Washington ranks first, followed by MIT and Boston Univ. Papers produced by Univ Cambridge have the highest citations, followed by Univ Michigan and Boston Univ. Alternatively, Univ Michigan ranks first in ACPP, followed by Stanford Univ and Univ Western Australia.

In addition, this paper analyzed the cooperation in disinformation research among the top 30 high-yield research institutions. As shown in Figure 4, each node represents an organization. The line between nodes represents the cooperation relationship between organizations. The thickness of the line indicates the frequency of cooperation. The thicker the line, the closer the cooperation relationship between the two.

It may be concluded that the top 30 institutions show close or sparse cooperation with each other. The higher-ranking institutions show stronger cooperation ability, and among them, Harvard Univ has the widest cooperation range. It has cooperated with 25 institutions in disinformation, published five relevant papers with MIT, cooperated with Yale Univ and Univ N Carolina four times, and cooperated with Univ Sydney, Univ Penn, and Univ California Irvine three times. The Univ N Carolina has partnerships with 21 institutions, such as the Univ Penn, the Univ Calif San Francisco, and Boston Univ. In the cooperative network, the relationship between Univ Bristol and Univ Western Australia is the most stable (*n* = 24), and the other strong cooperative relationships are Duke Univ and Univ N Carolina (*n* = 9), Boston Univ and Univ N Carolina (*n* = 5). However, some institutions such as Univ Toronto, Univ Washington, and Arizona State Univ have not cooperated with other top 30 institutions.

### 3.5. Keywords Analysis

By analyzing the keywords, we can understand the key fields of disinformation research. Therefore, this research conducted a statistical analysis on the keywords of 5666 papers. After cleaning the data, the visualized data are shown in Figure 5. They show the bubble chart of the top 30 high-frequency keywords over time, which uses three-dimensional data to explain the changing trend of disinformation research. The first dimension is time, which spans from 2002 to 2021. The second dimension is the total frequency of every keyword, and the higher the ranking, the higher the frequency. The third dimension is the frequency of some keywords in some years; the size of the bubble will change with the occurrence frequency of the keyword in that year.

“Misinformation” has the highest frequency (*n* = 817), followed by “social media” (*n* = 684) and “COVID-19” (*n* = 600). There are some differences in the research process of these keywords. They can be divided into three categories according to the characteristic of keywords with time variation. The first kind possesses continuous popularity, including “false memory”, “misinformation effect”, and “suggestibility”. The bubble chart of such keywords almost fills the entire time interval, and the bubbles are relatively large, without obvious changes over time. The second kind is those keywords produced in recent 10 years, such as “Machine Learning”, “Deep Learning”, “Natural Language Processing”, “Facebook”, and “infodemiology”. Such keywords indicate researchers have widely used intelligent algorithms to identify and classify disinformation on social media and study its mechanism [74,75,76]. The last kind is keywords of rising popularity, which will appear sooner or later. Since 2017, the size of bubbles belonging to this kind of keyword has shown a trend of linear growth. They are “misinformation”, “social media”, “fake news”, and “public health”.

### 3.6. Highly Cited Papers

There are 176 most cited papers among the 5666 papers in disinformation. The top 20 highly cited papers are listed in Table 4. Focusing on the distribution of these papers, 35% of the papers are jointly published by two or more countries, and 65% of the papers are from the USA.

This paragraph summarizes the research findings of some papers. In 2011, Acemoglu et al. discussed the possibility that media sources, politicians, and the state could manipulate misinformation [77]. In 2012, Lewandowsky et al. examined the mechanisms by which such misinformation is disseminated in society and pointed out that works of fiction are also the source of misinformation [78]. Public consumers may be harmed, misled, and disappointed by such misinformation [79]. Jolley et al. highlighted the potentially detrimental consequences of anti-vaccine conspiracy theories in 2014 [80]. In the same year, Nyhan et al. tested the effectiveness to reduce vaccine misperceptions of four interventions [81]. After the 2016 presidential election, the public has shown concerns about misinformation on social media [82]. Allcott and Gentzkow found social media was an important source of election news in 2016 [83]. Although there were many studies on the source and transmission mechanism of misinformation, there was still a lack of perfect measures to reduce the harm of misinformation. In 2020, Pennycook et al. showed participants were far worse at discerning between true and false content, and the level of true discernment in participants’ subsequent sharing intentions could be nearly tripled using a simple accuracy reminder [84].

### 3.7. Analysis of Hot Papers

Different from highly cited papers, hot papers represent the latest research directions. Among the 5666 papers, 16 hot papers are shown in Table 5. Among them, there are three review papers and 13 papers. From the time dimension, the 16 hot papers are concentrated in 2019, 2020 and 2021. At this junction, COVID-19 broke out and spread rapidly. From the spatial dimension, hot papers are distributed in four main countries, including seven in the USA, six in the UK, two in The Netherlands, and one in Germany.

These hot papers mainly represent two popular topics. Some papers focus on the negative influence of misinformation on social media during COVID-19. For instance, Gao et al. revealed the correlation between public mental health problems and misinformation exposure on social media [85]. Allington et al. proposed conspiracy theories are the barriers to controlling the epidemic. Particularly, four papers underlined the impact of vaccine misinformation in vaccine promotion [86,87,88,89]. The other papers introduced measures to deter misinformation. Chen et al. put forward a solution based on blockchain for fake news [90]. Erku et al. attached importance to pharmacists’ role in controlling misinformation [91]. Most of these hot papers were produced based on COVID-19, which proves that after the outbreak of COVID-19, scholars’ attention to disinformation has risen to a high level.

## 4. Discussion

A total of 5666 papers were identified for the present bibliometric assessment of research on disinformation in this study. The results of the study indicate that the number of papers produced increased the most from 2019 to 2021. This change indicates that the outbreak of COVID-19 has pushed disinformation research to a new climax and has considerably influenced the research orientation and hot areas of disinformation. Among many countries/regions, the USA leads the field of disinformation research, with the largest number of publications and the highest frequency of citations. This is attributed to the manipulation of disinformation in many political events, providing the best breeding context and sufficient cases for the USA to study disinformation, such as the Cold War between the USA and the Soviet Union and the 2016 US presidential election. However, it should be noted that the share of the USA in the world is decreasing year by year, which indicates that the significance of disinformation research is beginning to penetrate other countries/regions, such as Italy, The Netherlands, New Zealand, and other developed countries. Moreover, the research in this field in developing countries is not very brilliant. China ranks fourth in terms of number of publications; however, its collaboration with other countries is not deep and extensive. As the only Chinese author in the top 30, Zhu had no collaborations with other high-yield authors. The reasons for this are twofold. On one hand, it is attributed to geographical differences. Physical distance prevents Chinese authors from collaborating with other highly productive authors. On the other hand, there is the lag of academic research. Zhu has been studying rumor propagation since 2016 and has published 10 articles related to rumor propagation models since 2019. Starting later than other scholars, other Chinese authors working with Zhu have not yet been able to stand out. Therefore, developing countries, represented by China, still have a lot of room for development.

The most prolific contributors to the field of disinformation are Ecker and Lewandowsky. Lewandowsky engaged in research on the intrinsic link between disinformation dissemination and cognitive behavior in 2005, while Ecker began to be engaged in related research later. They have worked closely together and have jointly published highly cited articles such as “Misinformation and Its Correction: Continued Influence and Successful Debiasing” and “Beyond Misinformation: Understanding and Coping with the ‘Post-Truth’ Era”. During this period, they conducted numerous experiments to investigate the mechanisms of misinformation transmission in society so as to reveal the effects of misinformation in memory work and to outline options for dealing with misinformation in the post-truth era.

Based on the results of the previous keywords analysis, two turning points can be seen to have affected scholars’ research on the key areas of disinformation: the emergence of the world wide web with various social network sites and the outbreak of COVID-19, as shown in Table 6. Before the emergence of social networking sites, the misinformation effect was a hot issue in the field. The keywords with a high frequency of co-occurrence with misinformation included age difference, memory, children, adults, etc. During this period, scholars paid attention to the group heterogeneity of misinformation in memory [92,93,94,95]. Before the outbreak of COVID-19, the Internet was an important source of disinformation. Keywords with high co-occurrence frequency with social media included media, fake news, Facebook, and trust. Researchers were committed to studying the mechanism of disinformation in social networks [96,97]. After the outbreak of COVID-19, researchers have tended to explore the impact of disinformation on epidemic prevention and control and public health [98]. The high-frequency keywords coexisting with COVID-19 are vaccine, health, and medical treatment. In addition, at this stage, the topic of infodemic has become hot, with the focus on the mixture of false information and true information after the outbreak of infectious diseases [99,100]. Relevant research technologies include machine learning [101,102], deep learning [103], and complex networks [104].

Evaluating disinformation research from a historical perspective is critical for measuring the current and future impacts of disinformation. By scrutinizing key papers and analyzing information about the authors’ countries/regions, institutions, disciplines, and topics, we can present a portrait of misinformation research that will enable future scholars to evaluate the direction of research.

A comprehensive study of the countries/regions, institutions, publications, and authors that have contributed most to disinformation research reveals that disinformation research has long been centered in the USA and the UK. The centrality of the USA and the UK is reflected in the quantity of high-impact papers, institutions, authors, and academic journals. However, this dynamic is changing, with the rise of research powers from countries/regions such as China, Italy, and The Netherlands. If we bring the time from 2002 closer to the present, we can see that scholars from these countries/regions are increasingly occupying the ranks of high-impact authors in the sequence. This trend is particularly evident when we compare ESI Highly Cited Papers (with a 10-year statistical cycle) and ESI Hot Papers (with a 3-year statistical cycle). We can see that ESI highly cited papers mainly come from predominant countries such as the UK and the USA, but the first place among ESI hot papers has become the work of Chinese scholars. We can argue that these trends will continue and influence the subsequent disinformation research.

Another corroboration of the rise of emerging power is the rise of emerging academic journals. As we can see, journals such as the International Journal of Environmental Research and Public Health and Vaccines of MDPI Press are becoming more and more important platforms in disinformation research, and the territory of traditional publishers such as Elsevier and Springer is shrinking step by step.

Another trend in the future of disinformation research is that research forces are becoming increasingly diverse and the research vision will become broader. We can see that many of the highly productive authors in this field are not from major countries and institutions. At the same time, there is not a close collaborative chain among many authors. The analysis of the research areas further illustrates the broad field and disciplinary span of disinformation research. The 5666 papers involve 617 research fields, among which the most published papers belong to interdisciplinary fields such as public health, computer science, engineering, and policy. In addition, attention to the research of disinformation has also been paid in other interdisciplinary fields such as agricultural economy, food science, electrical engineering, and chemistry.

With regard to research topics, infodemic public health related to COVID-19 and the application of big data technology in disinformation are hot issues in disinformation research. In the past three years, the number of relevant papers has increased sharply, which is also confirmed by the increasing development trend of derived keywords such as infodemic and COVID-19 in keyword analysis.

Although disinformation research has achieved considerable success in many aspects, the existing research results indicate that more in-depth research should be conducted in the following directions to meet the crushing challenges posed by disinformation.

First, the typical formation process of disinformation has obvious stage-specific characteristics. How to sensitively perceive and distinguish the stages in which the disinformation is located and analyze the evolution mechanism of sub-stages is a topic that needs further research. This requires not only more involvement and extensive collaboration from scholars in the fields of information management, compute science, social management, and other disciplines, but perhaps also a new theory to provide theoretical support.

Second, according to the studies of scholars, social media—represented by Twitter—has become a breeding ground for disinformation. It is more meaningful to analyze the causes of disinformation from the perspective of social media users, such as user profiles of vulnerable groups in social media and behavioral tracking of high-impact users.

Finally, AI technologies facilitate the triggering and proliferation of disinformation, such as Deepfake and Botnets. It becomes crucial to make AI technologies serve the governance of disinformation. For example, optimization algorithms are used to establish social media information source-awareness mechanisms, detection algorithms are used to screen fake information, and algorithmic transparency is enhanced to improve users’ ability to distinguish information.

## 5. Conclusions

This study shows the research overview of disinformation from 2002 to 2021. Based on the bibliometric analysis, this paper shows the distribution of global disinformation research, analyzes the differences and connections among countries/regions, core authors, and research institutions, and outlines the four hot topics of disinformation, being group heterogeneity of misinformation in memory, disinformation mechanism in social media, public health related to COVID-19, and application of big data technology in disinformation. This study can help scholars in the field of information-related research quickly grasp the full picture of disinformation in global research, and help researchers understand the current research results in this field to carry out more in-depth research.

## Figures and Tables

**Figure 1 ijerph-19-16849-f001:**
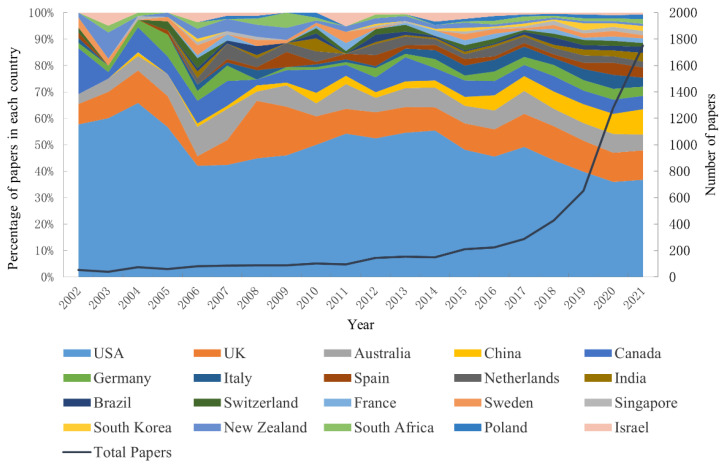
Number of yearly papers in the top 20 high-yield countries/regions.

**Figure 2 ijerph-19-16849-f002:**
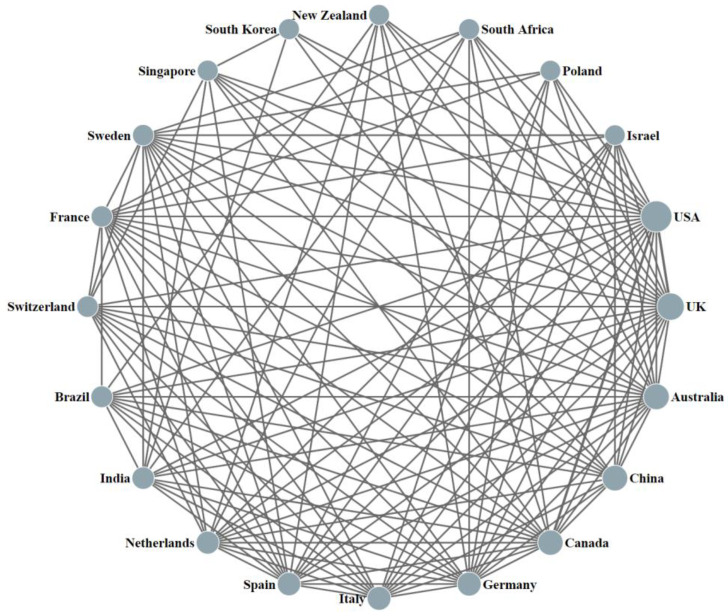
Cooperation network map of the top 20 high-yield countries/regions.

**Figure 3 ijerph-19-16849-f003:**
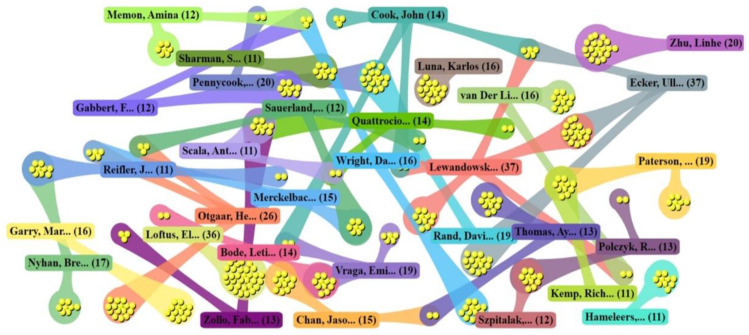
DDA cluster diagram of cooperation among the top 30 authors.

**Figure 4 ijerph-19-16849-f004:**
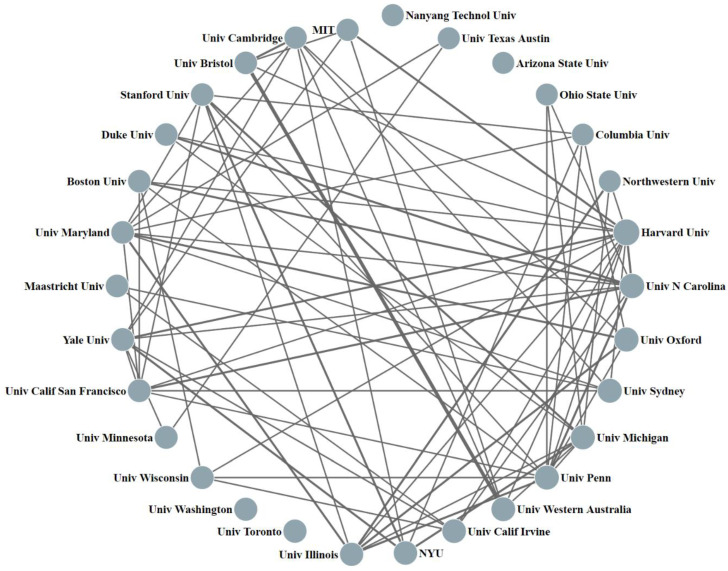
Cooperation cluster diagram of the top 30 institutions with the highest productivity.

**Figure 5 ijerph-19-16849-f005:**
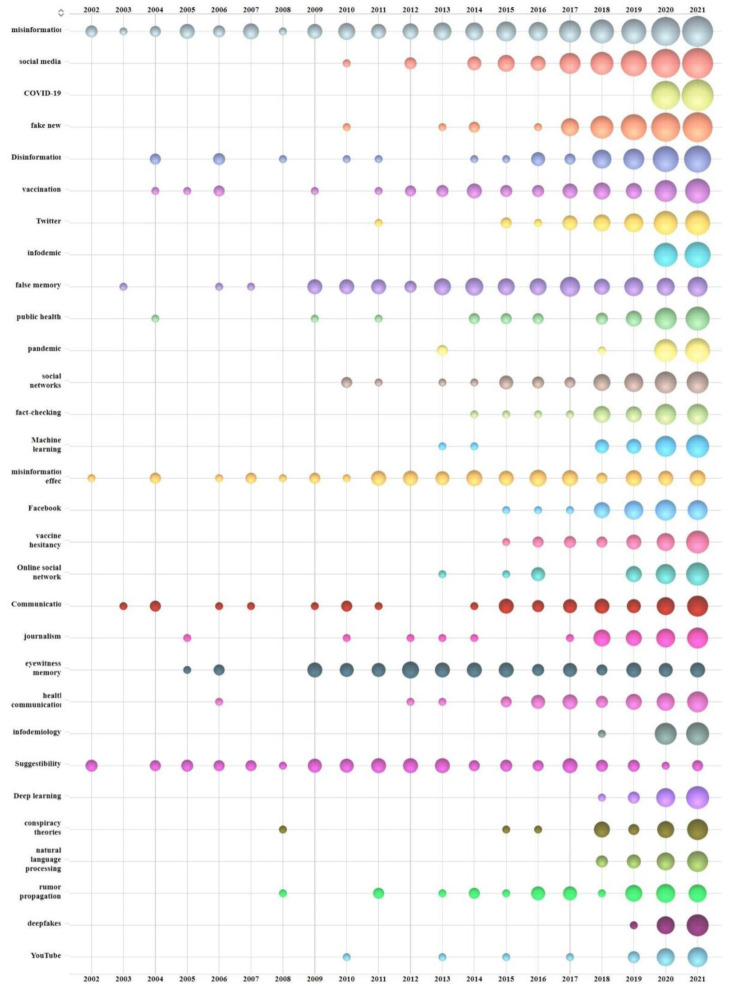
Annual variation bubble chart of disinformation research keywords.

**Table 1 ijerph-19-16849-t001:** The top 30 most attractive journals.

Rank	Publication	TP	Category	Countries/Regions	IF
1	*APPL COGNITIVE PSYCH*	118	Psychology, Experimental	USA	2.36
2	*PLOS ONE*	113	Multidisciplinary Sciences	USA	3.75
3	*J MED INTERNET RES*	102	Medical Informatics, Health Care Sciences & Services	Canada	7.08
4	*INT J ENV RES PUB HE*	83	Public, Environmental & Occupational Health	Switzerland	4.61
5	*MEMORY*	78	Psychology, Experimental	UK	2.52
6	*IEEE ACCESS*	62	Telecommunications, Engineering, Electrical &Electronic, Computer Science, Information Systems	USA	3.48
7	*SOC MEDIA SOC*	60	Communication	UK	4.65
8	*DIGIT JOURNAL*	45	Communication	UK	6.85
9	*NEW MEDIA SOC*	43	Communication	USA	5.31
10	*INT J COMMUN-US*	40	Communication	USA	1.64
11	*PHYSICA A*	40	Physics, Multidisciplinary	The Netherlands	3.78
12	*P NATL ACAD SCI USA*	38	Multidisciplinary Sciences	USA	12.78
13	*HEALTH COMMUN*	37	Communication, Health Policy & Services	USA	3.50
14	*MEM COGNITION*	36	Psychology, Experimental	USA	2.48
15	*FRONT PSYCHOL*	35	Psychology, Multidisciplinary	Switzerland	4.23
16	*INFORM COMMUN SOC*	35	Communication, Sociology	UK	5.05
17	*JOURNAL PRACT*	31	Communication	UK	2.33
18	*MEDIA COMMUN-LISBON*	31	Communication	Portugal	3.04
19	*VACCINE*	31	Medicine, Research & Experimental, Immunology	The Netherlands	4.17
20	*BMC PUBLIC HEALTH*	29	Public, Environmental & Occupational Health	UK	4.14
21	*VACCINES-BASEL*	28	Medicine, Research & Experimental, Immunology	Switzerland	4.96
22	*INFORM PROCESS MANAGE*	27	Information Science & Library Science	UK	7.47
23	*J APPL RES MEM COGN*	27	Psychology, Experimental	The Netherlands	4.6
24	*JOURNALISM*	26	Communication	USA	3.19
25	*SCI REP-UK*	26	Multidisciplinary Sciences	UK	4.99
26	*AM BEHAV SCI*	25	Psychology, Clinical, Social Sciences, Interdisciplinary	USA	2.53
27	*FRONT PUBLIC HEALTH*	25	Public, Environmental & Occupational Health	Switzerland	6.46
28	*JMIR PUBLIC HLTH SUR*	25	Public, Environmental & Occupational Health	Canada	4.11
29	*CONTRACEPTION*	24	Obstetrics & Gynecology	The Netherlands	3.05
30	*HUM VACC* *IMMUNOTHER*	24	Biotechnology & Applied Microbiology, Immunology	USA	4.53

**Table 2 ijerph-19-16849-t002:** The top 30 authors regarding disinformation research from 2002 to 2021.

Rank	Author	TP	TC	ACPP	H-Index	Institution	Countries/Region
1	Ecker, UKH	37	2468	66.70	18	Univ Western Australia	Australia
2	Lewandowsky, S	37	3063	82.78	21	Univ Western Australia	Australia
3	Loftus, EF	36	1477	41.03	17	Univ Calif Irvine	USA
4	Otgaar, H	26	249	9.58	9	Maastricht Univ	The Netherlands
5	Pennycook, G	20	1409	70.45	12	Univ Regina	Canada
6	Zhu, Linhe	20	228	11.40	9	Jiangsu Univ	China
7	Paterson, HM	19	337	17.74	8	Univ Sydney	Australia
8	Rand, DG	19	1409	74.16	12	MIT	USA
9	Vraga, EK	19	756	39.79	11	Univ Minnesota	USA
10	Nyhan, B	17	2269	133.47	12	Dartmouth Coll	USA
11	Garry, M	16	463	28.94	12	Victoria Univ Wellington	New Zealand
12	Luna, K	16	175	10.94	8	Univ Minho	Portugal
13	van der Linden, S	16	712	44.50	11	Univ Cambridge	UK
14	Wright, DB	16	1016	63.50	16	Florida Int Univ	USA
15	Chan, JCK	15	381	25.40	10	Iowa State Univ	USA
16	Merckelbach, H	15	217	14.47	9	Maastricht Univ	The Netherlands
17	Bode, L	14	581	41.50	8	Georgetown Univ	USA
18	Cook, J	14	2003	143.07	10	George Mason Univ	USA
19	Quattrociocchi, W	14	1494	106.71	11	Ca Foscari Univ Venice	Italy
20	Polczyk, R	13	36	2.77	4	Jagiellonian Univ	Poland
21	Thomas, AK	13	236	18.15	6	Tufts Univ	USA
22	Zollo, F	13	1250	96.15	10	Ca Foscari Univ Venice	Italy
23	Gabbert, F	12	956	79.67	11	Univ Portsmouth	USA
24	Memon, A	12	899	74.92	10	Univ Aberdeen	UK
25	Sauerland, M	12	78	6.50	5	Maastricht Univ	The Netherlands
26	Szpitalak, M	12	30	2.50	3	Jagiellonian Univ	Poland
27	Hameleers, M	11	111	10.09	5	Univ Amsterdam	The Netherlands
28	Kemp, RI	11	298	27.09	8	UNSW Sydney	Australia
29	Reifler, J	11	2092	190.18	9	Univ Exeter	UK
30	Scala, A	11	1430	130.00	11	CNR ISC	Italy

**Table 3 ijerph-19-16849-t003:** The top 30 institutions with the highest productivity.

Rank	Institutions	TP	TC	ACPP	Countries/Regions
1	Univ Washington	235	4761	20.26	USA
2	MIT	228	4392	19.26	USA
3	Boston Univ	178	4867	27.34	USA
4	Univ Cambridge	164	7053	43.01	UK
5	Univ N Carolina	154	2349	15.25	USA
6	Harvard Univ	143	4723	33.03	USA
7	Columbia Univ	129	2029	15.73	USA
8	Univ Penn	124	4472	36.06	USA
9	Univ Sydney	121	2134	17.64	Australia
10	Univ Michigan	113	6647	58.82	USA
11	Univ Oxford	83	996	12	UK
12	NYU	83	3084	37.16	USA
13	Univ Western Australia	76	3416	44.95	Australia
14	Univ Toronto	73	2413	33.05	Canada
15	Univ Wisconsin	73	1854	25.4	USA
16	Univ Illinois	72	2050	28.47	USA
17	Yale Univ	62	1938	31.26	USA
18	Univ Calif Irvine	60	1952	32.53	USA
19	Univ Bristol	57	1853	32.51	UK
20	Univ Maryland	56	664	11.86	USA
21	Univ Minnesota	55	695	12.64	USA
22	Duke Univ	55	1193	21.69	USA
23	Univ Calif San Francisco	54	603	11.17	USA
24	Maastricht Univ	50	511	10.22	The Netherlands
25	Stanford Univ	49	2298	46.9	USA
26	Northwestern Univ	48	804	16.75	USA
27	Nanyang Technol Univ	46	1108	24.09	Singapore
28	Univ Texas Austin	45	572	12.71	USA
29	Arizona State Univ	44	643	14.61	USA
30	Ohio State Univ	43	923	21.47	USA

**Table 4 ijerph-19-16849-t004:** Top 20 papers highly cited in disinformation from 2002 to 2021.

Rank	Title	Keywords	Journal	TC	Countries/Regions
1	Social Media and Fake News in the 2016 Election	partisan bias, polarization, online, accuracy, beliefs, impact	*J ECON PERSPECT*	1043	USA
2	Misinformation and Its Correction: Continued Influence and Successful Debiasing	misinformation, false beliefs, memory updating, debiasing	*PSYCHOL SCI PUBL INT*	921	USA, Australia
3	Why do humans reason? Arguments for an argumentative theory	argumentation, confirmation bias, decision making, dual process theory, evolutionary psychology, motivated reasoning, reason-based choice, reasoning	*BEHAV BRAIN SCI*	806	USA, France
4	Opioid Epidemic in the United States	opioid abuse, opioid misuse, nonmedical use of psychotherapeutic drugs, nonmedical use of opioids, national survey on drug use and health, opioid guidelines	*PAIN* *PHYSICIAN*	673	USA
5	The spreading of misinformation online	misinformation, virality, Facebook, rumor spreading, cascades	*P NATL ACAD SCI USA*	604	USA, Italy
6	Effective Messages in Vaccine Promotion: A Randomized Trial	vaccines, myths, rumor, autism, false, misperceptions, misinformation	*PEDIATRICS*	583	USA
7	DEFINING FAKE NEWS, A typology of scholarly definitions	facts, fake news, false news, misinformation, news, parody, satire	*DIGIT* *JOURNAL*	502	Singapore
8	Anti-vaccine activists, Web 2.0, and the postmodern paradigm—An overview of tactics and tropes used online by the anti-vaccination movement	anti-vaccination, health communication, internet, postmodernism, vaccines, web 2.0	*VACCINE*	444	Canada
9	Mind the Hype: A Critical Evaluation and Prescriptive Agenda for Research on Mindfulness and Meditation	mindfulness, meditation, psychotherapy, neuroimaging, contemplative science, adverse effects, media hype, misinformation	*PERSPECT PSYCHOL SCI*	440	Australia, The Netherlands, USA
10	Mental health problems and social media exposure during COVID-19 outbreak		*PLOS ONE*	413	China
11	Attitudes to vaccination: A critical review	Europe, vaccination, immunization, public health, choice, attitude, perception, hesitancy	*SOC SCI MED*	364	UK
12	The Effects of Anti-Vaccine Conspiracy Theories on Vaccination Intentions	continued influence, African Americans, beliefs, attitudes, misinformation, HIV/aids, impact, online	*PLOS ONE*	362	UK
13	Vaccine hesitancy: the next challenge in the fight against COVID-19	COVID-19, SARS-CoV-2 vaccine, vaccine hesitancy, healthcare staff, vaccine safety, Israel	*EUR J* *EPIDEMIOL*	338	Israel
14	Beyond Misinformation: Understanding and Coping with the Post-Truth Era	misinformation, fake news, post-truth politics, demagoguery	*J APPL RES MEM COGN*	327	UK, Australia, USA
15	Opinion Dynamics and Learning in Social Networks	Bayesian updating, consensus, disagreement, learning, misinformation, non-Bayesian models, rule of thumb behavior, social networks	*DYN GAMES APPL*	320	USA
16	Lazy, not biased: Susceptibility to partisan fake news is better explained by lack of reasoning than by motivated reasoning	fake news, news media, social media, analytic thinking, cognitive reflection test, intuition, dual process theory	*COGNITION*	306	USA
17	Fake news on Twitter during the 2016 US presidential election		*SCIENCE*	284	USA
18	Fighting COVID-19 Misinformation on social media: Experimental Evidence for a Scalable Accuracy-Nudge Intervention	social media, decision making, policy making, reflectiveness, social cognition, open data, open materials, preregistered	*PSYCHOL SCI*	283	Canada, USA
19	Motivational pathways to STEM career choices: Using expectancy-value perspective to understand individual and gender differences in STEM fields	career choices, stem, individual and gender differences, expectancy-value theory	*DEV REV*	268	USA
20	NASA Faked the Moon Landing-Therefore, (Climate) Science Is a Hoax: An Anatomy of the Motivated Rejection of Science	scientific communication, policymaking, climate science	*PSYCHOL SCI*	268	Australia, Switzerland

**Table 5 ijerph-19-16849-t005:** Hot papers in disinformation research from 2002 to 2021.

Rank	Title	Keywords	Journal	TC	Countries/Regions
1	Mental health problems and social media exposure during COVID-19 outbreak		*PLOS ONE*	413	China
2	Vaccine hesitancy: the next challenge in the fight against COVID-19	COVID-19, SARS-CoV-2 vaccine, Vaccine hesitancy, Healthcare staff, Vaccine safety, Israel	*EUR J* *EPIDEMIOL*	338	Israel
3	Fighting COVID-19 misinformation on social media: experimental evidence for a scalable accuracy-nudge intervention	social media, decision making, policy making, reflectiveness, social cognition, open data, open materials, preregistered	*PSYCHOL SCI*	283	Canada, USA
4	Systematic literature review on the spread of health-related misinformation on social media	Misinformation, Fake news, Health, Social media	*SOC SCI MED*	244	UK, Italy
5	A comprehensive review of the COVID-19 pandemic and the role of IoT, Drones, Ai, Blockchain, and 5G in managing its impact	Coronavirus, COVID-19, pandemic, transmission stages, global economic impact, UAVs for disaster management, Blockchain, IoMT applications, IoT, AI, 5G	*IEEE* *ACCESS*	219	India, Qatar
6	The digital transformation of innovation and entrepreneurship: Progress, challenges, and key themes	Digital transformation, Innovation, Entrepreneurship, Digital innovation, Digital platforms, Openness, Generativity, Affordance	*RES* *POLICY*	191	USA, UK
7	Health-protective behavior, social media usage, and conspiracy belief during the COVID-19 public health emergency	Conspiracy beliefs, COVID-19, health-protective behaviors, public health, social media	*PSYCHOL MED*	168	UK
8	Transmission of SARS-CoV-2: a review of viral, host, and environmental factors	attack rate, infections	*ANN* *INTERN MED*	155	USA
9	Conspiracy theories as barriers to controlling the spread of COVID-19 in the US	Conspiracy theories, COVID-19, Prevention, Vaccination, Political ideology, Media use, Vaccination misinformation	*SOC SCI MED*	152	USA
10	Social media and vaccine hesitancy	vaccines	*BMJ GLOB HEALTH*	108	USA, South Africa
11	Measuring the impact of COVID-19 vaccine misinformation on vaccination intent in the UK and USA	public-health, hesitancy, exposure, opinion, news	*NAT HUM BEHAV*	107	USA, UK, Belgium
12	Considering emotion in COVID-19 vaccine communication: addressing vaccine hesitancy and fostering vaccine confidence	fear, misinformation, metanalysis, appeals	*HEALTH COMMUN*	104	USA
13	A survey on fake news and rumor detection techniques	Fake news, Rumors, Natural language processing, Data mining, Text mining, Classification, Machine learning, Deep learning	*INFORM SCIENCES*	90	Italy
14	Fact-checking as risk communication: the multi-layered risk of misinformation in times of COVID-19	Risk communication, misinformation, trust, uncertainty	*J RISK RES*	79	USA, Germany
15	An incentive-aware blockchain-based solution for internet of fake media things	Blockchain, Fake news, Internet of fake media things, Proof-of-authority	*INFORM PROCESS MANAG*	57	Canada, Taiwan (China), USA, Kuwait
16	When fear and misinformation go viral: Pharmacists’ role in deterring medication misinformation during the ‘infodemic’ surrounding COVID-19	Coronavirus, Misinformation, COVID-19, Pandemics, Pharmacists	*RES SOC ADMIN PHARM*	55	Australia, Ethiopia

**Table 6 ijerph-19-16849-t006:** Stages of misinformation research.

Stage	Keywords	Theme
Before the emergence of WWW	misinformation, age-difference, recall, memory, children, adults, suggestibility, judgment, memory conformity, false memory	group heterogeneity of misinformation in memory
After the emergence of social networking sites and before the outbreak of COVID-19	social media, media, fake news, communication, Facebook, credibility, trust, bias, disinformation, journalism, truth	disinformation mechanism in social media
After the outbreak of COVID-19	COVID-19, vaccination, health, management, knowledge, prevalence, pandemic	public health related to the COVID-19
	infodemic, coronavirus, infodemiology, crisis, infoveillance, Twitter, machine learning, deep learning, social networks, fake news detection, diffusion, fact-checking, blogs, dynamics	application of big data technology in infodemic

## Data Availability

Not applicable.

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
