# Peer review of "Disinformation: A Bibliometric Review"

_ijerph, 2022, doi:10.3390/ijerph192416849_

Round 1
Reviewer 1 Report
This article reveals a very interesting research regarding the theme of disinformation associated with research. The introduction to the topic reveals the care in the scientific evidence on the keywords used associated with prototypical varieties, as well as, the perspectives, in the scope, social, economic and in the individual perspective. The Materials and Methods are very well explained and structured, explaining step by step their bibliometric analysis. It should be noted that the analysis carried out reveals very interesting information, as well as comparisons that are very valuable for future research. The methodology and analysis methods were well explained and care was taken to present the main objectives of each analysis.
Author Response
Response to comments from reviewer #1
Comment 1:
This article reveals very interesting research regarding the theme of disinformation associated with research. The introduction to the topic reveals the care in the scientific evidence on the keywords used associated with prototypical varieties, as well as, the perspectives, in the scope, social, economic and in the individual perspective. The Materials and Methods are very well explained and structured, explaining step by step their bibliometric analysis. It should be noted that the analysis carried out reveals very interesting information, as well as comparisons that are very valuable for future research. The methodology and analysis methods were well explained and care was taken to present the main objectives of each analysis.
Response to comment 1:
Thank you for reading this paper out of your busy schedule and commenting on our work. We are encouraged by your recognition. Following your suggestion, we included more discussion of the future frontiers of disinformation in the manuscript.

Reviewer 2 Report
Dear authors,
I was pleased to review your manuscript. It certainly is an interesting angle of observing the topic of disinformation. There are quite a few studies so far that use the same method in portraying the fields of disinformation, misinformation and-or fake information. Therefore, I fail to recognize your added value.
In general, the topic is greatly saturated and well-established. Still, I did not find your proper reasoning and logic for performing bibliometric analysis. This greatly erodes the positioning and overall merit of your work.
Next, you tend to simply reproduce what is already indicated on figures and vice versa. Basically, not much-added value here either. We need you to tell us a story behind your findings, not repeat what we already saw-read. Also, some figures are not needed and if you ask me, should be removed (e.g., Figure 3)
Maybe the greatest setback is that you do not provide future prospects or critically recognize gaps that scholars could attend to in the future. One of the outcomes of bibliometric analysis is to provide future directions, to clearly identify research fronts, and to pinpoint places that need extensions. This is missing.
Taken all together, I humbly believe that you should familiarize yourself more with the bibliometric analysis and reflect on some good practices - and only then restart work on your manuscript.
Unfortunately, in its current state, your manuscript does not add much value to the research field and there are some serious flaws in the positioning and operationalization of the study itself.
Best of luck with your research.
Author Response
Response to comments from reviewer #2
Comment 2.1:
I was pleased to review your manuscript. It certainly is an interesting angle of observing the topic of disinformation. There are quite a few studies so far that use the same method in portraying the fields of disinformation, misinformation and-or fake information. Therefore, I fail to recognize your added value.
Response to comment 2.1:
Thank you for your sincere comments and for acknowledging the perspective of this study. According to your comment, we have conducted a survey in the manuscript to compare with similar papers. Undeniably, there are a number of current bibliometric studies related to disinformation and its related concepts. However, the research content and scope of these articles differ significantly from this paper. For example, "BIBLIOMETRIC ANALYSIS OF MEDIA DISINFORMATION AND FAKE NEWS IN SOCIAL NETWORKS" focuses on identifying trends in the literature related to fake news and media disinformation in social networks, identifying changes in the number of articles posted based on the time dimension. Similarly, the article "Bibliometric analysis of rumor detection via web of science from 1989 to 2021" focuses on the growth trends of the literature related to rumors and predicts future growth. The article entitled "Infodemic, Misinformation and Disinformation in Pandemics: Scientific Landscape and the Road Ahead for Public Health Informatics Research" focuses on information epidemics since the outbreak of epidemics and summarizes the future direction of information epidemics. Compared to these articles, focusing on disinformation itself and summarizing its research hotspots and future gaps in terms of literature sources, research subjects, and research results is the greatest added value of this article. Therefore, it can provide reference value for relevant researchers to grasp the overall situation of the field and understand the research hotspots.
Comment 2.2:
In general, the topic is greatly saturated and well-established. Still, I did not find your proper reasoning and logic for performing bibliometric analysis. This greatly erodes the positioning and overall merit of your work.
Response to comment 2.2:
Your comments are extremely valuable. We have revised the paper in accordance with your suggestions. The revisions are as follows.
A total of 5666 papers were identified for the present bibliometric assessment of research on disinformation in this study. The results of the study indicate that the number of papers produced increased the most from 2019 to 2021. This change indicates that the outbreak of COVID-19 has pushed disinformation research to a new climax and has considerably influenced the research orientation and hot areas of disinformation. Among many countries/regions, the USA leads the field of disinformation research, with the largest number of publications and the highest frequency of citations. This is attributed to the manipulation of disinformation in many political events, providing the best breeding context and sufficient cases for the U.S. to study disinformation, such as the U.S. Soviet Cold War, the 2016 U.S. presidential election. China is ranked 4 in terms of number of publications, however, collaboration with other countries is not deep and extensive. As the only Chinese author in the top 30, Zhu did not generate collaborations with other high yield authors. The reasons for this are twofold. On the one hand, it is attributed to geographical differences. Physical distance prevents Chinese authors from collaborating with other highly productive authors. On the other hand, there is the lag of academic research. Zhu has been studying rumor propagation since 2016, and has published ten articles related to rumor propagation models since 2019. Starting later than other scholars, other Chinese authors working with Zhu have not yet been able to stand out.
The most prolific contributors to the field of disinformation are Ecker and Lewandowsky. Lewandowsky engaged in research on the intrinsic link between disinformation dissemination and cognitive behavior in 2005, while Ecker engaged in research later. They have worked closely together and have jointly published highly cited articles such as "Misinformation and Its Correction: Continued Influence and Successful Debiasing" and " Beyond Misinformation: Understanding and Coping with the ‘Post-Truth’ Era". During this period, they conducted numerous experiments to investigate the mechanisms of misinformation transmission in society, to reveal the effects of misinformation in memory work, and to outline options for dealing with misinformation in the post-truth era.
At the same time, we believe that the application of bibliometric methods in this topic is reasonable. The study of this topic involves rich concepts, many disciplines, a large number of scholars and a huge amount of literature, and bibliometric visualization methods can reveal the current status of research and future expectations of this topic more comprehensively.
Comment 2.3:
Next, you tend to simply reproduce what is already indicated on figures and vice versa. Basically, not much-added value here either. We need you to tell us a story behind your findings, not repeat what we already saw-read.
Response to comment 2.3:
Thank you for your valuable suggestions. We have revised the paper in accordance with your suggestions. But the bibliometric analysis is based on the literature. Therefore, in the results section a visualizable analysis of current disinformation research is performed using tools and relevant factual statements are made based on graphs as well as tables. In the Discussion section, we present a blended analysis and tell the story behind the findings. This structure makes the article more organized, and we have seen some excellent papers with a similar structure, such as "Artificial intelligence in the AEC industry: Scientometric analysis and visualization of research activities", which was published in Automation in Construction (rank first in civil engineering) and has been cited 143 times.
Comment 2.4:
Also, some figures are not needed and if you ask me, should be removed (e.g., Figure 3).
Response to comment 2.4:
Thank you for your suggestions. It has been redacted according to your comment.
Comment 2.5:
Maybe the greatest setback is that you do not provide future prospects or critically recognize gaps that scholars could attend to in the future. One of the outcomes of bibliometric analysis is to provide future directions, to clearly identify research fronts, and to pinpoint places that need extensions. This is missing.
Response to comment 2.5:
Thank you very much for your suggestion. The frontiers in the paper are not specifically summarized, but are scattered throughout the rest of the article. In response to your suggestion, the paper has been revised to include a look at the future frontiers of disinformation. Based on the analysis in the previous papers and the discussion of relevant experts, the paper ends with an indication of the gaps that scholars can focus on in the future. The details are as follows.
The participation of many scholars and institutions has led to the completion and maturation of relevant research in the field of disinformation. However, there are still gaps in many aspects, and in-depth research can be conducted in the following areas in the future to address the heavy challenges posed by disinformation.
First of all, the typical formation process of disinformation has obvious stage-specific characteristics. How to sensitively perceive and distinguish the stages in which the information fog is located and analyze the evolution mechanism of sub-stages is a topic that needs further research.
Second, social media is a breeding ground for disinformation. It is more meaningful to analyze the causes of information fog from the perspective of social media users, such as user profiles of vulnerable groups in social media and behavioral tracking of high-impact users.
Finally, AI technologies facilitate the triggering and proliferation of disinformation, such as Deepfake as well as Botnets. It becomes crucial to make AI technologies serve the governance of disinformation. For example, the use of optimization algorithms to establish social media information source-awareness mechanisms, the use of detection algorithms to screen fake information, and enhanced algorithmic transparency to improve users' ability to distinguish information.
Comment 2.6:
Taken all together, I humbly believe that you should familiarize yourself more with the bibliometric analysis and reflect on some good practices - and only then restart work on your manuscript. Unfortunately, in its current state, your manuscript does not add much value to the research field and there are some serious flaws in the positioning and operationalization of the study itself.
Response to comment 2.6:
Thank you for your practical advice. We are honored that you read our articles carefully in the midst of your busy schedule. We read a large number of excellent articles that use bibliometric methods and summarized their general practices. We have reflected on the shortcomings of our article and reorganized the ideas of our article. As you suggested, we have revised our paper and reorganized Introduction section, Discussion section, and Results section. In particular, in the Discussion section, we consulted experts in the field to discuss the frontiers and research gaps in the field of disinformation. We hope that the above work will lead to an improvement of the manuscript.

Reviewer 3 Report
The review paper uses the Derwent Data Analyzer (DDA) to analyze the disinformation publications published between 2002 and 2021 based on the Web of Science. It is a very good written paper with a very clear structure and very good organized content. I suggest the acceptance of this manuscript.
Some minor problems that don’t affect the overall quality of the manuscript. However, fixing these problems will help the paper to reach a higher standard.
1. In the author list, it seems to be an extra “and *” at the end.
2. Figure 1 seems not to be centered.
3. The width of Table 5 can be enlarged a little bit so the entire table will become shorter.
4. There is an extra space after the sentence “…., and medical misinformation trends and themes[59]”. A similar situation happens at the end of the sentence “such as VOSviewer, Citespace, and Derwent Data Analyzer (DDA).” Another example with extra space can be found in the middle of the sentence “n the same year, Nyhan et al. tested the effectiveness to reduce vaccine misperceptions of 4 interventions[69].”
Author Response
Response to comments from reviewer #3
Comment 3.1:
The review paper uses the Derwent Data Analyzer (DDA) to analyze the disinformation publications published between 2002 and 2021 based on the Web of Science. It is a very good written paper with a very clear structure and very good organized content. I suggest the acceptance of this manuscript. Some minor problems that don’t affect the overall quality of the manuscript. However, fixing these problems will help the paper to reach a higher standard.
In the author list, it seems to be an extra “and *” at the end.
Response to comment 3.1:
Thank you very much for your recognition of this article. Your comment is a recognition of the efforts made by our co-authors. Because of your approval, our confidence grows and we are hopeful for the future of this manuscript. We are sorry for such a mistake. We have corrected it according to what you pointed out and set the correct subscript.
Comment 3.2:
- Figure 1 seems not to be centered.
Response to comment 3.2:
Thanks to you for spotting this problem. We have set centering for Figure 1.
Comment 3.3:
The width of Table 5 can be enlarged a little bit so the entire table will become shorter.
Response to comment 3.3:
Thank you very much for your comment. We have verified the dimensions of Table 5 and found that they do not match the dimensions of the other tables. Based on your suggestion, we have stretched the width of Table 5 and tried to align it with the other tables as much as possible.
Comment 3.4:
There is an extra space after the sentence “…., and medical misinformation trends and themes[59]”. A similar situation happens at the end of the sentence “such as VOSviewer, Citespace, and Derwent Data Analyzer (DDA).” Another example with extra space can be found in the middle of the sentence “n the same year, Nyhan et al. tested the effectiveness to reduce vaccine misperceptions of 4 interventions[69].”
Response to comment 3.4:
Thank you for your advice. We are in awe of your good eyesight. There was indeed a redundant space in " such as VOSviewer, Citespace, and Derwent Data Analyzer (DDA)." and " n the same year, Nyhan et al. tested the effectiveness to reduce vaccine misperceptions of 4 interventions[69].", which we have done to remove. There is no extra space after the sentence "…., and medical misinformation trends and themes[59]", although it is distracting. So we searched for the cause and found that it was an inconsistency in the font, and have now made the change.

Reviewer 4 Report
The abstract failed to explain the objective and the problem that will be addressed in the research.
Some information contained in the introduction could be addressed in a section entitled disinformation (which is the main theme of the article).
In the conclusion, it was necessary to explain the future works based on the theme proposed in the research.
Author Response
Response to comments from reviewer #4
Comment 4.1:
The abstract failed to explain the objective and the problem that will be addressed in the research.
Response to comment 4.1:
Thank you for your constructive comments. Based on your suggestion, we have reorganized this section by referring to the structured abstract in the IJERPH journal as follows.
Objectives: This paper aimed to provide a systematic review of relevant articles from the perspectives of literature distribution, research hotspots and existing results to obtain the frontier directions in the field of disinformation. Methods: We analyzed disinformation publications published between 2002 and 2021 using bibliometric methods based on the Web of Science. There were 5,666 papers analyzed using Derwent Data Analyzer (DDA). Results: The result shows that the USA is the most influential country in this area, while Ecker and Lewandowsky from the University of Western Australia published the largest volumes of papers. Keywords such as "social media", "COVID-19" and "vaccination" have gained immense popularity recently. Conclusions: We summarized four themes that are most concerned by scholars: group heterogeneity of misinformation in memory, disinformation mechanism in social media, public health related to COVID-19, and application of big data technology in the infodemic. The future agenda of disinformation is summarized from three aspects: the mechanism of disinformation, social media users and the application of algorithms. This work can be a meaningful resource for researchers’ study in the area of disinformation.
Comment 4.2:
Some information contained in the introduction could be addressed in a section entitled disinformation (which is the main theme of the article).
Response to comment 4.2:
Thank you very much for your suggestion. Your suggestion will indeed lead to a clearer and more focused structure of the article. However, IJERPH imposes restrictions on the typographic structure of the article, and this article is structured based on the following requirements. https://www.mdpi.com/journal/ijerph/instructions. If you feel it is necessary, we will discuss with the editor to make adjustments as you suggest. Thank you very much.
Research Manuscript Sections
Introduction: The introduction should briefly place the study in a broad context and highlight why it is important. It should define the purpose of the work and its significance, including specific hypotheses being tested. The current state of the research field should be reviewed carefully and key publications cited. Please highlight controversial and diverging hypotheses when necessary. Finally, briefly mention the main aim of the work and highlight the main conclusions. Keep the introduction comprehensible to scientists working outside the topic of the paper.
Materials and Methods: They should be described with sufficient detail to allow others to replicate and build on published results. New methods and protocols should be described in detail while well-established methods can be briefly described and appropriately cited. Give the name and version of any software used and make clear whether computer code used is available. Include any pre-registration codes.
Results: Provide a concise and precise description of the experimental results, their interpretation as well as the experimental conclusions that can be drawn.
Discussion: Authors should discuss the results and how they can be interpreted in perspective of previous studies and of the working hypotheses. The findings and their implications should be discussed in the broadest context possible and limitations of the work highlighted. Future research directions may also be mentioned. This section may be combined with Results.
Conclusions: This section is mandatory.
Comment 4.3:
In the conclusion, it was necessary to explain the future works based on the theme proposed in the research.
Response to comment 4.3:
Thank you very much for your suggestion. We apologize that the article did omit some relevant statements about the frontiers of disinformation. In response to your suggestion, the article has been expanded to include an outlook on the future frontiers of disinformation. Based on the analysis in the previous article and the discussion of relevant experts, the article ends with an indication of the gaps that scholars can focus on in the future. The details are as follows.
The participation of many scholars and institutions has led to the completion and maturation of relevant research in the field of disinformation. However, there are still gaps in many aspects, and in-depth research can be conducted in the following areas in the future to address the heavy challenges posed by disinformation.
First of all, the typical formation process of disinformation has obvious stage-specific characteristics. How to sensitively perceive and distinguish the stages in which the information fog is located and analyze the evolution mechanism of sub-stages is a topic that needs further research.
Second, social media is a breeding ground for disinformation. It is more meaningful to analyze the causes of information fog from the perspective of social media users, such as user profiles of vulnerable groups in social media and behavioral tracking of high-impact users.
Finally, AI technologies facilitate the triggering and proliferation of disinformation, such as Deepfake as well as Botnets. It becomes crucial to make AI technologies serve the governance of disinformation. For example, the use of optimization algorithms to establish social media information source-awareness mechanisms, the use of detection algorithms to screen fake information, and enhanced algorithmic transparency to improve users' ability to distinguish information.
Round 2
Reviewer 2 Report
Dear authors,
Thank you for your sending the revised version of your manuscript.
Although somewhat improved, I do not find it robust enough and with a clear added value. I am sorry for this discouraging feedback, but the bibliometric analysis per se is very limited unless you really go in-depth and capture multiple angles. Not only capture, but elaborate and analyze.
For your reference, I am disclosing several robust bibliometric analyses that you could have referred to (please mind that I do not direct you to any of my contributions).
https://www.sciencedirect.com/science/article/pii/S0148296319305727?casa_token=G7-7kDAqPL8AAAAA:CYrRGSeIpx9OQlH0Ol9wJzHp4TgRqCgp2F35RVorZupJzTB3NmeamlBmaCv9IOInlrHoNCnukHQ
https://www.sciencedirect.com/science/article/pii/S0148296321004951?casa_token=Moh1EJjBIAIAAAAA:DjvMvIdRVpuUocWtlCh-rBe6jYXtYFAGZ5_K6UTyns-4Eb1vzy341qRYhW2-CGzNWPeeVDH_l_c
https://www.sciencedirect.com/science/article/pii/S0959652619347559?casa_token=B-WvDbGjukAAAAAA:lDJdp6g_o9jl1gfeV5S7zX99SEilWuK3VeHBG7tTmOZRJMbaJmc8W6p4G5Ln5b_earPybsWBUKI
https://www.sciencedirect.com/science/article/pii/S0148296321008213
Author Response
Response to comments from reviewer #2
Comment 2.1:
Thank you for your sending the revised version of your manuscript.
Although somewhat improved, I do not find it robust enough and with a clear added value. I am sorry for this discouraging feedback, but the bibliometric analysis per se is very limited unless you really go in-depth and capture multiple angles. Not only capture, but elaborate and analyze.
For your reference, I am disclosing several robust bibliometric analyses that you could have referred to (please mind that I do not direct you to any of my contributions).
https://www.sciencedirect.com/science/article/pii/S0148296319305727?casa_token=G7-7kDAqPL8AAAAA:CYrRGSeIpx9OQlH0Ol9wJzHp4TgRqCgp2F35RVorZupJzTB3NmeamlBmaCv9IOInlrHoNCnukHQ
https://www.sciencedirect.com/science/article/pii/S0148296321004951?casa_token=Moh1EJjBIAIAAAAA:DjvMvIdRVpuUocWtlCh-rBe6jYXtYFAGZ5_K6UTyns-4Eb1vzy341qRYhW2-CGzNWPeeVDH_l_c
https://www.sciencedirect.com/science/article/pii/S0959652619347559?casa_token=B-WvDbGjukAAAAAA:lDJdp6g_o9jl1gfeV5S7zX99SEilWuK3VeHBG7tTmOZRJMbaJmc8W6p4G5Ln5b_earPybsWBUKI
https://www.sciencedirect.com/science/article/pii/S0148296321008213
Response to comment 2.1:
Thank you to the insightful recommendation. Your suggestion is very helpful in improving the quality of the paper.
Guided by the reviewer, we have carefully read several high quality papers and have revised the manuscript substantially and are very grateful for the reviewer's help. "Past, present, and future of customer engagement" (10.1016/j.jbusres.2021.11.014) revealed the major trends in article, author, country, and journal performance, as well as the past, present, and future thematic trends of CE research. "Mapping the electronic word-of-mouth (eWOM) research: A systematic review and bibliometric analysis" (10.1016/j.jbusres.2021.07.015) used bibliographic coupling to identify the four primary topics at the forefront of research on eWOM following Andersen (2019).
These papers all provide a vision of the future of the research topic based on the past and the present. Their research is comprehensive and in-depth, and we have carefully studied their writing style and revised the manuscript in light of their methodology. Similar to their approach, our study is also based on the analysis of existing literature to arrive at the full picture of disinformation research and the analysis of the thematic features of disinformation research to find the frontier issues, but their work is really more in-depth and provides us with samples to improve the quality of our paper. However, unlike Donthu's approach following Andersen (2019), our approach is based on hotspots, thematic emergence, and other aspects to find frontiers. Inspired by these papers, we can learn more from them and make more innovations in our next paper. We would like to thank the reviewer again for his/her help.
Our revisions to the manuscript are as follows:
(1)In Section Discussion
Evaluating disinformation research from a historical perspective is critical for measuring the current and future impacts of disinformation. By scrutinizing key papers and analyzing information about the authors' countries/regions, institutions, disciplines, and topics, we can present a portrait of misinformation research that will enable future scholars to evaluate the direction of research.
A comprehensive study of the countries/regions, institutions, publications, and authors that have contributed most to disinformation research reveals that disinformation research has long been centered in the USA and the UK. The centrality of the USA and the UK is reflected in the quantity of high-impact papers, institutions, authors, and academic journals. However, this dynamic is changing, with the rise of research powers from countries/regions such as China, Italy, and the Netherlands. If we bring the time from 2002 closer to the present, we can see that scholars from these countries/regions are increasingly occupying the ranks of high-impact authors in the sequence. This trend is particularly evident when we compare ESI Highly Cited Papers (with a 10-year statistical cycle) and ESI Hot Papers (with a 3-year statistical cycle). We can see that ESI highly cited papers mainly come from predominantly countries such as the UK and the USA, but the first place among ESI hot papers has become the work of Chinese scholars. We can argue that these trends will continue and influence the future of disinformation research in the upcoming -time.
Another corroboration of the rise of emerging power is the rise of emerging academic journals. As we can see, journals such as International Journal of Environmental Research and Public Health and Vaccines of MDPI Press are becoming more and more important platforms in disinformation research, and the territory of traditional publishers such as Elsevier and Springer is shrinking step by step.
Another trend in the future of disinformation research is that research forces are becoming more diverse and the research vision will become broader. We can see that many of the highly productive authors in this field are not from major countries and institutions. At the same time, there is not a close collaborative chain among many authors. The analysis of the research areas further illustrates the broad field and disciplinary span of disinformation research. The 5,666 papers involve 617 research fields, among which the most published papers belong to interdisciplinary fields, such as public health, computer science, engineering, and policy. In addition, as for the research of disinformation, attention has also been paid to in other interdisciplinary fields, such as agricultural economy, food science, electrical engineering, and chemistry.
With regard to research topics, infodemic public health related to COVID-19, and the application of big data technology in disinformation are hot issues in disinformation research. In the recent three years, the number of relevant papers has increased sharply, which is also confirmed by the increasing development trend of derived keywords such as infodemic and COVID-19 in keyword analysis.
Although disinformation research has achieved considerable success in many aspects, the existing research results indicate that more in-depth research should be conducted in the following directions to meet the crushing challenges posed by disinformation.
First of all, the typical formation process of disinformation has obvious stage-specific characteristics. How to sensitively perceive and distinguish the stages in which the disinformation is located and analyze the evolution mechanism of sub-stages is a topic that needs further research. This requires not only more involvement and extensive collaboration from scholars in the fields of information management, compute science, social management, and other disciplines, but perhaps also a new theory to provide theoretical support.
Second, according to the studies of scholars, social media, represented by Twitter, has become a breeding ground for disinformation. It is more meaningful to analyze the causes of disinformation from the perspective of social media users, such as user profiles of vulnerable groups in social media and behavioral tracking of high-impact users.
(2)In Section Materials and Methods
Bibliometric analysis usually consists of two parts: performance analysis and science mapping[68]. Performance analysis contributes to help discover emerging themes and recent advances in a field, the influence of leading scholars, and the impact of different journals and schools of thought[69].This paper use performance analysis to find leading countries/regions and journals, as well as prolific authors and institutions.
Bibliometric analysis relies on citation and co-citation analysis for quantitative review[70]. Citations indicate the use of a specific work by a citing scholar and reveal the value, importance, and influence of that work[69]. In this analysis, the most cited works are used to illuminate the theoretical underpinnings, methodologies, and key themes that drive the discipline in the field of disinformation.
Keyword co-occurrence analysis is a bibliometrics method, which assumes that when two keywords appear in multiple articles at the same time, there must be some correlation between the concepts reflected[71]. It is considered appropriate to express central themes in some fields using keywords[72]. Based on the large number of literatures, keyword co-occurrence analysis is used to identify the central topics in disinformation.
(3)In Section Introduction
There are a few papers that provide a bibliometric review of the concepts related to disinformation. They provide an overview in some perspectives. Lee used bibliometric methods to reveal the evolution of academic networks in the field of misinformation, but his study was limited to the period 2009-2018, which clearly did not reflect the latest developments[59]. Some scholars reviewed rumors, fake news, and information epidemics, but these are related concepts of disinformation, narrowing the scope of what disinformation can accommodate[60-62]. Tito et al. and Yeung et al. 's review limited disinformation to the social media[63] and medical fields[63, 64], respectively, and although these two fields are the main sites of disinformation, which limits the grasp of the filed in its entirety. Patra et al.’s review summarized existing results comprehensively to some extent, but rarely tap into research hotspots and research trends that are directly related to disinformation[65].
(4)Some minor revision